# Taxonomy of *Pseudomonas* spp. determines interactions with *Bacillus subtilis*

Mark Lyng,[1] Birta Þórisdóttir,[1] Sigrún H. Sveinsdóttir,[1] Morten L. Hansen,[2] Lars Jelsbak,[2] Gergely Maróti,[3] Ákos T. Kovács[1,4]

**ABSTRACT** Bacilli and pseudomonads are among the most well-studied microorganisms commonly found in soil and frequently co-isolated. Isolates from these two genera are frequently used as plant beneficial microorganisms; therefore, their interaction in the plant rhizosphere is relevant for agricultural applications. Despite this, no systematic approach has been employed to assess the coexistence of members from these genera. Here, we screened 720 fluorescent soil isolates for their effects on *Bacillus subtilis* pellicle formation in two types of media and found a predictor for interaction outcome in *Pseudomonas* taxonomy. Interactions were context-dependent, and both medium composition and culture conditions strongly influenced interactions. Negative interactions were associated with *Pseudomonas capeferrum*, *Pseudomonas entomophila*, and *Pseudomonas protegens*, and 2,4-diacetylphloroglucinol was confirmed as a strong (but not exclusive) inhibitor of *B. subtilis*. Non-inhibiting strains were closely related to *Pseudomonas trivialis* and *Pseudomonas lini*. Using such a non-inhibiting isolate, *Pseudomonas* P9_31, which increased *B. subtilis* pellicle formation demonstrated that the two species were spatially segregated in cocultures. Our study is the first one to propose an overall negative outcome from pairwise interactions between *B. subtilis* and fluorescent pseudomonads; hence, cocultures comprising members from these groups are likely to require additional microorganisms for coexistence.

**IMPORTANCE** There is a strong interest in the microbial ecology field to predict interaction among microorganisms, whether two microbial isolates will promote each other's growth or compete for resources. Numerous studies have been performed based on surveying the available literature or testing phylogenetically diverse sets of species in synthetic communities. Here, a high throughput screening has been performed using 720 *Pseudomonas* isolates, and their impact on the biofilm formation of *Bacillus subtilis* was tested. The aim was to determine whether a majority of *Pseudomonas* will promote or inhibit the biofilms of *B. subtilis* in the co-cultures. This study reports that *Pseudomonas* taxonomy is a good predictor of interaction outcome, and only a minority of *Pseudomonas* isolates promote *Bacillus* biofilm establishment.

**KEYWORDS** *Bacillus subtilis*, *Pseudomonas*, phylogenetics, biofilm, competition, secondary metabolite

Plant growth-promoting rhizobacteria (PGPR) possess great potential to replace traditional fertilizers and pesticides as a more sustainable alternative. In particular, isolates from *Bacillus* and *Pseudomonas* genera have been studied extensively due to their abilities to inhibit plant pathogens, induce plant systemic resistance, increase the growth rate of plants, and alleviate environmental stress (*Bacillus* reviewed in reference 1 and *Pseudomonas* reviewed in reference 2). Several studies combined isolates from the two genera and observed a synergistic increase in a trait of interest, be it plant growth or protection (3–7). However, few studies have investigated the cause of synergy; hence,

Address correspondence to Ákos T. Kovács, a.t.kovacs@biology.leidenuniv.nl.

The authors declare no conflict of interest.

See the funding table on p. 15.

little information is available on the pairwise compatibility of *Bacillus* and *Pseudomonas*, and whether environmental isolates engage in antagonism or mutualism.

*Bacillus subtilis* is one of the most well-studied model organisms in biology, serving as a prototypical example of biofilm formation and plant root colonization. *In vitro* biofilm development of *B. subtilis*, for example, the formation of highly structured colonies on agar-solidified medium and floating pellicle biofilms on the air-medium interface has been correlated with plant protection against pathogen in pot experiments and rhizosphere colonization in hydroponic settings (8–11). Several strains have demonstrated antagonism toward other microorganisms, especially plant pathogenic fungi (12). Such interactions are mainly mediated by secreted bioactive secondary metabolites, of which *B. subtilis* produces a diverse arsenal. Of note is plipastatin that shows antifungal properties (13, 14), while surfactin and bacillaene display more broad antimicrobial properties (15, 16). Positive interactions between *Bacillus* and *Pseudomonas* have also been observed, such as intraspecies division of labor (17) and interspecies cross-feeding (18).

*Pseudomonas* is a genus of numerous species comprising at least 166 types of strains that are found in soil, plant rhizospheres, marine habitats, and animal hosts (19–22). The impact of *Pseudomonas* spp. on human society ranges from human and plant pathogenicity to bioremediation, biocontrol, and biostimulation (23–26). Interestingly, there is a fine line between pathogenic and beneficial *Pseudomonas* spp. that mainly depends on the arsenal of secondary metabolites produced by a given strain (24, 27). Similarly, compatibility with *Bacillus* spp. also seems to rely mainly on a relatively small collection of secondary metabolites and defense mechanisms (28). Understanding the frequency of positive and negative interactions between *Bacillus* and diverse *Pseudomonas* spp., as well as the underlying mechanisms of such interactions would be of great interest to agricultural biotechnology relying on mixed consortia of species from the two genera.

Here, we cocultured 720 soil isolates selected for fluorescent *Pseudomonas* properties with the undomesticated type-strain *B. subtilis* DK1042 (hereafter DK1042) under floating biofilm (pellicle)-inducing conditions and used a spinning disc-based medium-throughput screen to quantify DK1042 biofilms. Using the two types of media, we demonstrate that although positive interactions with superior pellicle abundance were rare in both types of media, interactions were highly medium-dependent. By taxonomically characterizing isolates at the species level, we determined that DK1042 was least likely to be compatible with species closely related to the type of strains of *Pseudomonas capeferrum*, *Pseudomonas entomophila*, and *Pseudomonas protegens*, and most likely to be compatible with *Pseudomonas lini* and *Pseudomonas trivialis*. We found that *Pseudomonas* antagonism toward DK1042 correlates with the presence of specific biosynthetic gene cluster (BGC) encoding the synthesis of 2,4-diactylphloroglucinol (DAPG), which is a major inhibitor of DK1042 in colonies though not an exclusive defector in pellicles.

## RESULTS

### Pseudomonads negatively affect *B. subtilis*

Relationships between *B. subtilis* and fluorescent pseudomonads (i.e., soil-dwelling *Pseudomonas* sp., that are generally isolated based on their auto fluorescence derived from secondary metabolites, including pyoverdine or pyochelin) range from antagonism to co-existence and, potentially, to synergistic growth, but no systematic investigation of interaction outcomes between the two has been performed.

To address this, we performed a pairwise coculture of DK1042 and each of 720 *Pseudomonas* soil isolates in 96-well plates and measured DK1042 biomass using an Opera High-Content screening spinning disc platform (Fig. 1a). We screened the library against DK1042 *amyE*::$P_{hyperspank}$-*mKate2* constitutively expressing the fluorophore mKate2 and recorded the three-dimensional volume occupied by *Bacillus* as a proxy for cell abundance. In this work, we choose to concentrate on a single *B. subtilis* strain, DK1042 that is the most frequently exploited undomesticated strain of this species to study biofilm development (17, 29–38), and interaction with host plants and diverse

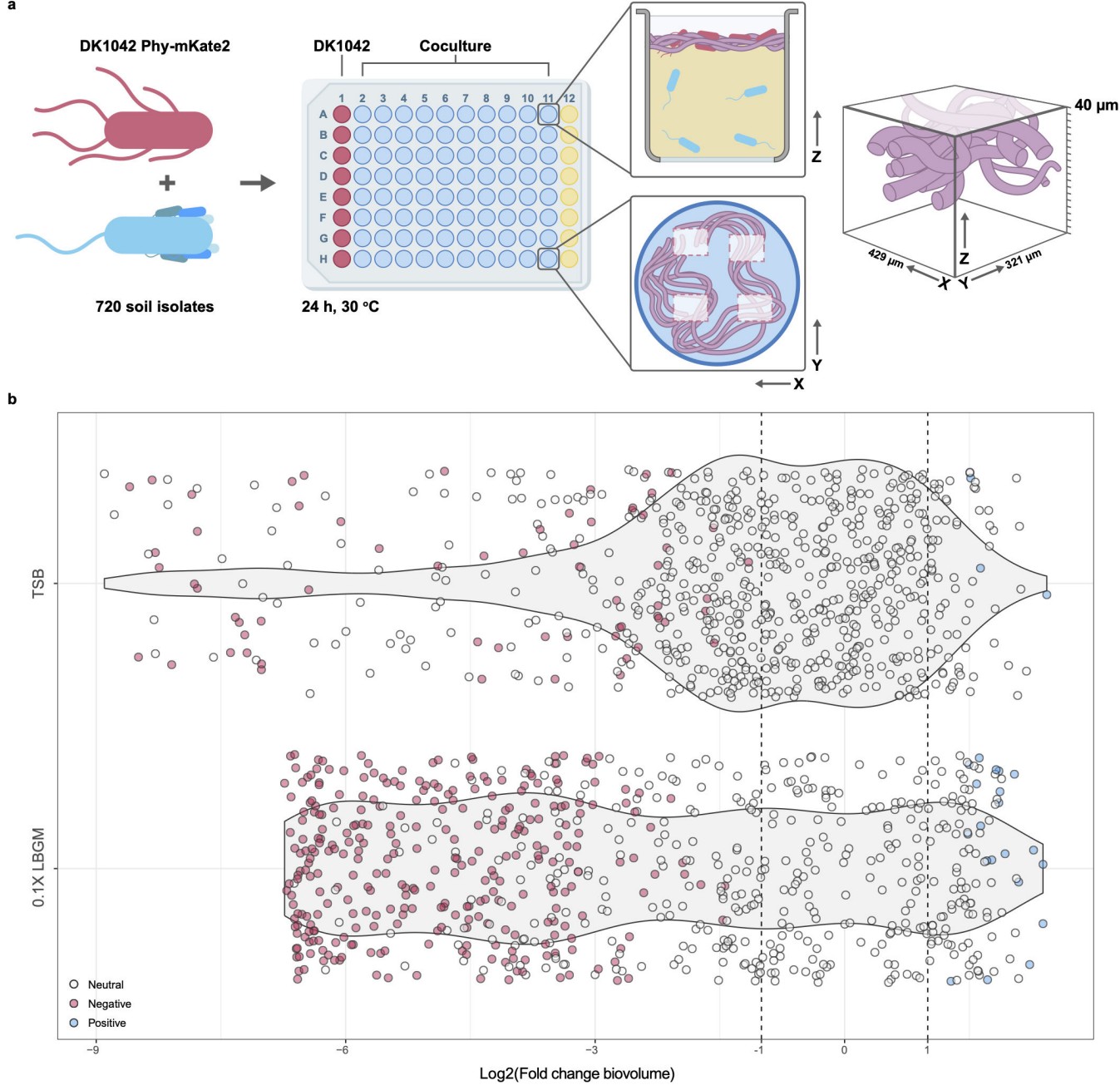

**FIG 1** Pseudomonads negatively affect *B. subtilis*. (a) DK1042 constitutively expressing mKate2 was mixed pairwise with 720 fluorescent soil isolates and cultured for 24 h at 30℃ in a microplate format. Monocultures of DK1042 (purple wells) were used for the comparison of pellicle formation, and wells with non-inoculated medium (yellow wells) were used to control for contamination. Each well was imaged in four positions with a Perkin Elmer Opera QEHS, acquiring images in the *Z*-direction every 2 µm to obtain a cube with height 40 µm. Media were removed before microscopy. (b) DK1042 biovolumes were compared between co- and monoculture to yield $\log_2$(fold change). Points represent medians of three replicates. Neutral, negative, and positive categories were assigned based on median and interquartile range.

microorganisms (10, 11, 39–44). Further, the *B. subtilis* group species form robust pellicle on the air-medium interface (18, 45, 46) that facilitated the high throughput screen. Each *Pseudomonas* isolate was cocultured three times with DK1042 both in rich Tryptic Soy Broth (TSB) medium and diluted biofilm-inducing lysogeny broth medium supplemented with glycerol and manganese (0.1× LBGM), and biovolume in coculture was compared with that in monocultures as $\log_2$(fold change) (Fig. 1b; Fig. S1). Rich medium

has been previously utilized to test the pellicle biofilm formation of *B. subtilis* group species (9, 18, 47), while diluted rich medium has been previously exploited to observe the induction of biofilm development produced by various secondary metabolites (38, 43).

The distribution of $\log_2$(FC) differed substantially between media types, suggesting that many neutral interactions in rich media become negative in diluted media, in line with stronger resource competition in nutrient-poor media among microorganisms (48). Examining the biovolume for each isolate (Fig. S2a) revealed that isolates from site P9 were more likely to antagonize DK1042 while isolates from the P8 site more often resulted in a high DK1042 biovolume. This also differed between media types, as cocultures in TSB had a higher degree of within-sample variance in biovolume measurements and therefore also in calculated $\log_2$(FC) values (Fig. S2b and c). We divided each isolate into three categories based on median $\log_2$(FC) and found that most interactions (regardless of medium) were neutral (Fig. 2a), while only a very small number of isolates had a positive effect on DK1042 (Fig. 1b). These results indicate that fluorescent pseudomonads very rarely stimulate *B. subtilis* cell abundance in TSB or 0.1× LBGM, and that inhibition of *B. subtilis* by *Pseudomonas* spp. is strongly influenced by medium composition.

## *Pseudomonas* taxonomy predicts interaction outcome

To determine the taxonomic distribution of isolates, we sequenced amplicons of the *rpoD* gene from the pools of isolates corresponding to their screening category (neutral, negative, and positive; Fig. 2; Data set S1).

The species diversity of each category demonstrates how the categorization based on *B. subtilis* pellicle formation selectively distributes specific phyla into separate categories (permutational analysis of variance, $P < 0.001$, Fig. 2d). The total library of 720 isolates was mainly comprised of *P. capeferrum*, *Pseudomonas helmanticensis*, *Pseudomonas koreensis*, and *P. protegens* (Fig. 2c; Table 1). A similar distribution occurred in the neutral category, while negative and positive strains differed significantly in species abundance. The pool of negative isolates contained more strains associated with the *Pseudomonas putida* group compared with the total library, but no single species was significantly depleted from this category. Differential abundance analysis revealed that the pool of positive isolates contained fewer isolates from the *P. putida* group, and the *P. protegens*, *Pseudomonas corrugata*, and *P. koreensis* subgroups, compared with the total library (Table 1). In contrast, *P. trivialis* and members of the *Pseudomonas mandelii* subgroup were enriched. As the pool of positive strains contained only 20 isolates, the statistical power in determining differential abundance was quite low. Therefore, caution should be applied when inferring from statistical tests in this group; hence, we refer to isolates henceforth as "inhibiting" (negative) or "non-inhibiting" (neutral and positive).

To test the predictions resulting from screening, we obtained four *Pseudomonas* strains closely related to species that were implicated in different interaction patterns, and cocultured them with DK1042 in liquid and solidified 1× LBGM and 0.1× LBGM media (Fig. 3). As predicted, *P. capeferrum* and *P. protegens* both inhibited pellicle formation (even in 1× LBGM), while neither *P. lini* nor *Pseudomonas poae* (in place of *P. trivialis*) reduced pellicle abundance or winkle formation (as measured via pixel standard deviation; Fig. 3b). However, on solid media, no pseudomonad was able to reduce the colony size of DK1042 on 1× LBGM, but on diluted 0.1× LBGM *P. protegens* strongly antagonized DK1042 (Fig. 3c). Thus, *Pseudomonas* antagonism of DK1042 depends not only on medium constituents but also on the mode of growth.

## Biosynthetic gene cluster abundance does not determine interaction outcome

*Pseudomonas* secondary metabolites are often implicated in *Bacillus-Pseudomonas* antagonism (16, 49, 50); therefore, we sequenced the genomes of 13 candidate isolates and predicted BGCs and BGC subgroups using antiSMASH (Fig. 4). Interestingly, there

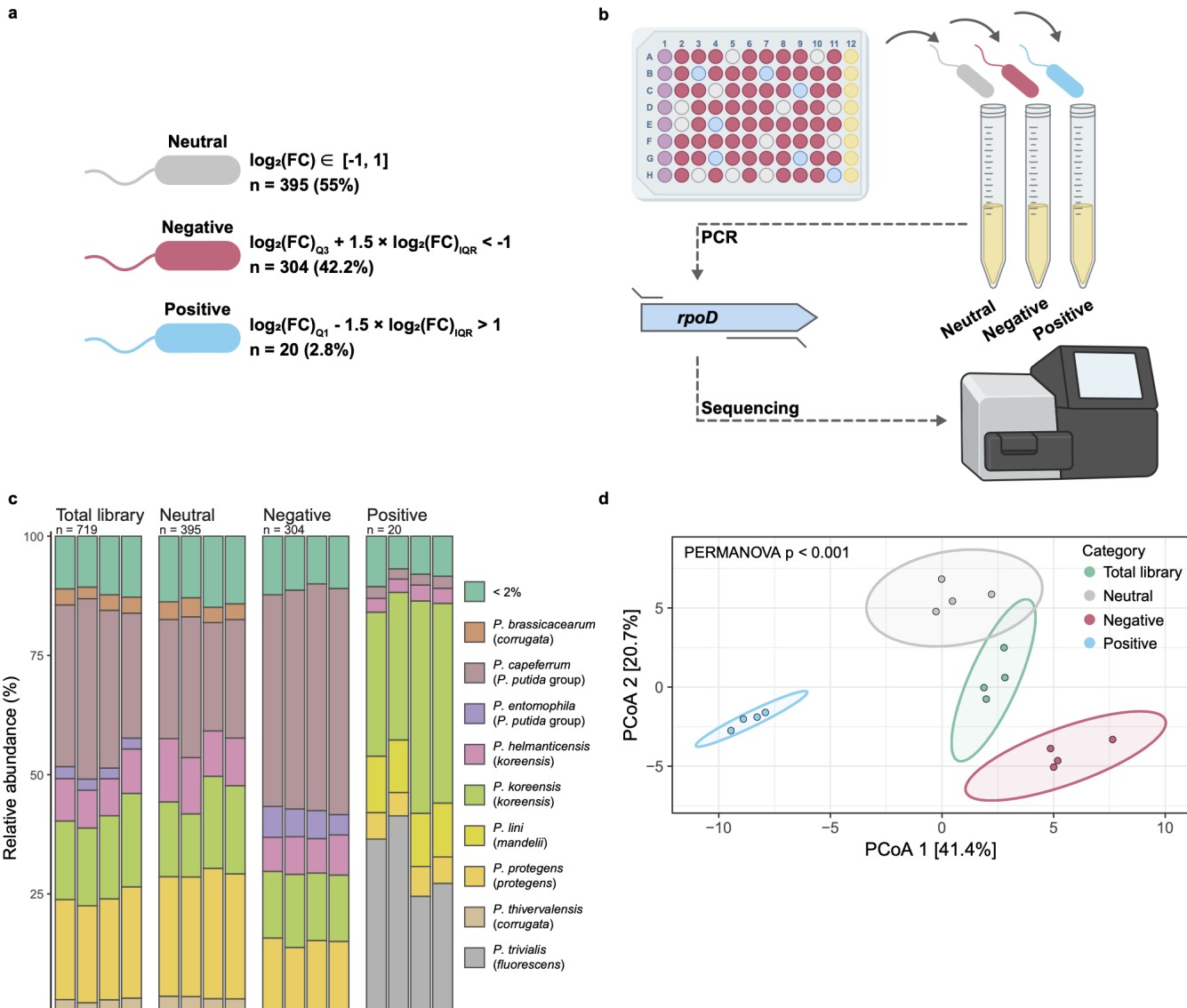

**FIG 2** Screening categories enriched for species-specific taxa. (a) Fluorescent isolates were assigned a category based on median and interquartile range, such that negatives resulted in DK1042 $\log_2(FC) < -1$ and positives in $\log_2(FC) > 1$. $n$ and percentages are from screening in 0.1× LBGM. (b) Isolates were grown in precultures and pooled in equal cell numbers according to their categorization to taxonomically characterize each category at the species level via *rpoD* amplicon sequencing. (c) Relative abundance of *Pseudomonas* spp. in each category. Parentheses indicate groups or *P. fluorescens* subgroups (see Table 1 for details on taxonomy). Taxons comprising <2% of a pool were merged. (d) Principal coordinate analysis of *rpoD* targeted amplicon sequencing. Each replicate is plotted with the corresponding 95% confidence interval ellipse. A permutational analysis of variance (PERMANOVA) shows a significant difference in sample type clustering.

was no significant difference in the abundance of encoded BGCs between inhibiting and non-inhibiting isolates (Fig. 4a). Therefore, we examined each genome for the presence of BGCs with known products (Fig. 4b). As expected, all isolates carried a version of pyoverdine, a fluorescent siderophore, but many non-inhibitory isolates were negative for most other BGCs compared with inhibiting isolates. Isolates from *P. mandelii* and *Pseudomonas jesenii* subgroups collectively encoded only one BGC with a predicted product (rhizoxin, originally isolated from *Paraburkholderia rhizoxinica* [51]). In contrast, colony-inhibiting isolates from both *P. corrugata* and *P. protegens* subgroups carried genes that encode 2,4-diacetylphloroglucinol (DAPG)-producing enzymes, while *P.*

**TABLE 1** Significantly differentially abundant species in screening categories compared with the total library

| *fluorescens* subgroup[a] | *Pseudomonas* sp. | Neutral (log$_2$FC) | Negative (log$_2$FC) | Positive (log$_2$FC) |
|---|---|---|---|---|
| asplenii | P. asplenii | 1.363[c] | −0.235 | −0.153 |
| corrugata | P. thivervalensis | −0.028 | −0.090 | −0.921[c] |
| | P. brassicacearum | −0.066 | −0.263 | −1.123[c] |
| fluorescens | **P. trivialis** | **−0.292** | **−0.381** | **2.900**[d] |
| | P. cedrina | −1.607[c] | 0.646 | −1.110 |
| | P. libanensis | −0.084 | −0.038 | −1.124[c] |
| | P. lurida | 0.153 | −1.097 | −1.196[c] |
| gesaardii | P. brenneri | −1.176 | 1.048 | −1.261[c] |
| koreensis | P. helmanticensis | 0.063 | 0.173 | −0.884[c] |
| | P. granadensis | 0.341 | −1.736[c] | −1.134 |
| | P. moraviensis | 0.031 | −0.267 | −1.244[c] |
| mandelii | **P. lini** | **−0.048** | **−0.425** | **2.623**[d] |
| | P. migulae | −0.510 | −0.348 | 1.829[c] |
| | P. frederiksbergensis | −0.163 | −1.292 | 1.574[d] |
| | P. mandelii | 0.088 | 1.007[c] | 0.396 |
| protegens | **P. protegens** | **−0.015** | **−0.090** | **−1.215**[d] |
| P. putida[b] | P. mosselii | −0.935 | 1.213[d] | −0.830 |
| | P. soli | −1.558[c] | 0.856 | −1.443[c] |
| | P. entomophila | −2.064[d] | 1.138[c] | −2.294[d] |
| | **P. capeferrum** | **−0.452** | **0.627** | **−2.493**[d] |

[a]As described in reference 19.
[b]Not a subgroup of *P. fluorescens*.
[c]Differentially abundant from the total library with FDR-adjusted *P*-value < 0.05.
[d]FDR-adjusted *P*-value < 0.01. Bold indicates predictions.

*protegens* additionally encoded enzymes for producing pyochelin, orfamide A, pyrrolnitrin, and pyoluteorin.

## Non-inhibiting isolates are spatially segregated from DK1042

While the main biofilm mode of *B. subtilis* in liquid culture involves growth at the air-liquid interface (30, 45), *Pseudomonas* spp. are usually observed colonizing submerged surfaces in laboratory environments, although examples of *Pseudomonas* pellicle formation do exist (53, 54). The presence of pellicles in cocultures with non-inhibiting isolates, therefore, suggests three potential scenarios: either non-inhibiting pseudomonads are outcompeted by *B. subtilis*, are growing with *B. subtilis* in the pellicle, or are spatially isolated from *B. subtilis* on the submerged surface of the microtiter plates. To determine which of these scenarios occurred with DK1042, we fluorescently labelled a non-inhibitor (P9_31), cocultured it statically with DK1042 in liquid 1× LBGM and 0.1× LBGM, and imaged the entire depth of the well (Fig. 5). In both media, P9_31 was spatially segregated to the submerged surface, even when DK1042 was mostly present in the pellicle. Interestingly, while cultivation in 1× LBGM resulted in a pellicle both with and without P9_31, cultivation in 0.1× LBGM required P9_31 for pellicle formation, even though the *Pseudomonas* isolate was present at the bottom of the well and not in the pellicle.

## DAPG is a biomarker for *B. subtilis* inhibition

Four of five colony-inhibiting isolates were found to contain the conserved *phl* BGC operon known to produce DAPG (55). DAPG production is conserved in specific phylogenetic groups of fluorescent pseudomonads, and the expression of the corresponding BGC is controlled by environmental factors and secondary metabolites produced by *Pseudomonas* (56–59). From 61 cocultures on solid 0.1× LBGM, 11 isolates

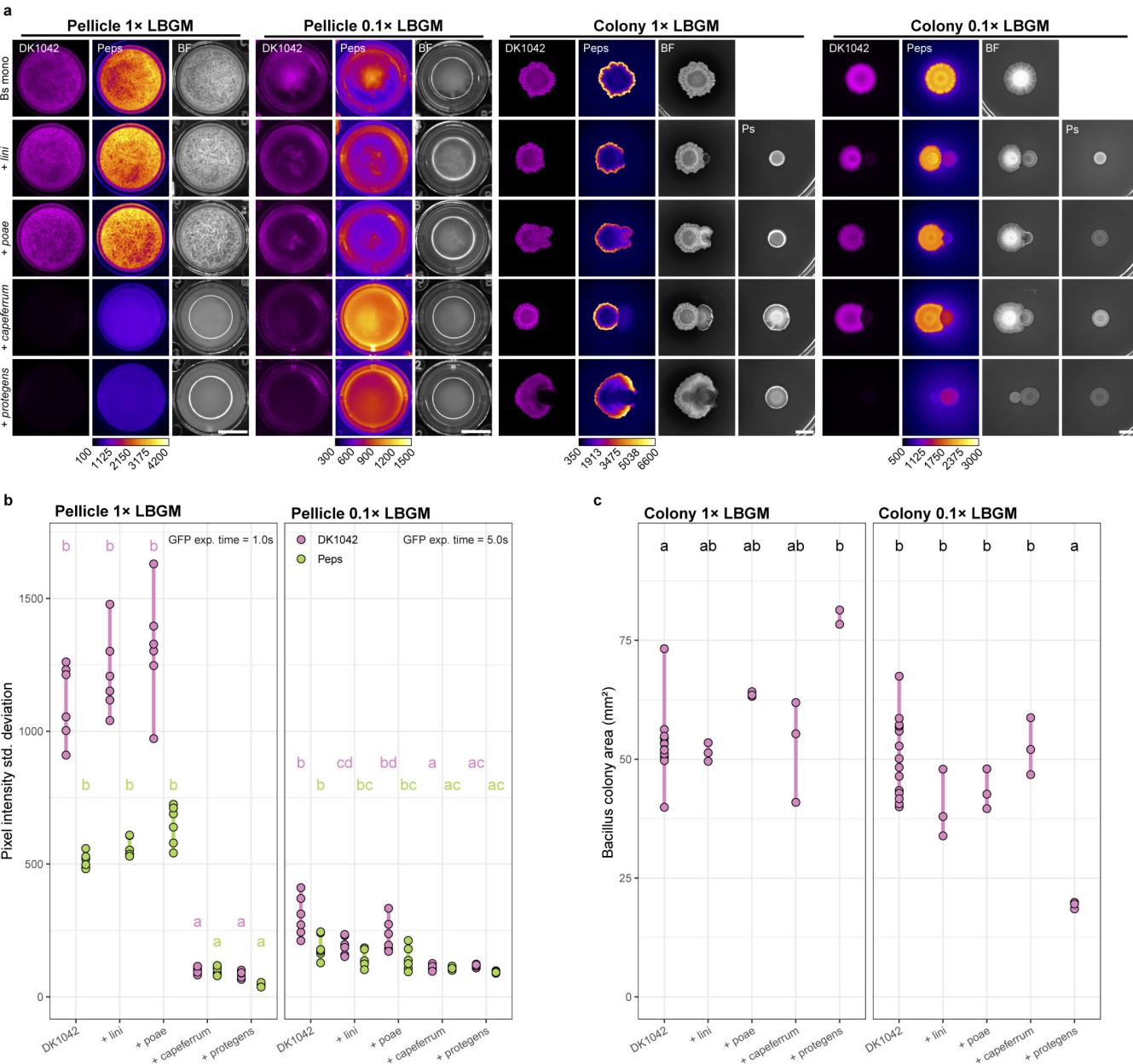

**FIG 3** Screening prediction of inhibitors and non-inhibitors. (a) Cocultures of DK1042 with four reference strains that were independent from the soil isolate library were used to assess the validity of interaction prediction resulting from screening *Pseudomonas* soil isolates. Cultures were grown at 30°C for 24 h (pellicles) or 72 h (colonies). Cultures were prepared with DK1042 reporting expression of the *epsA-O* operon using GFP. Scale bar = 5 mm. (b) Pellicle wrinkle formation represented by the standard deviation in pixel intensity (i.e., high standard deviation equals stronger wrinkle formation). *n* = 6 independent experiments. (c) Colony area of DK1042 spotted as monoculture or neighboring the four *Pseudomonas* reference strains. *n* = 3 independent cocultures. Grouping letters are from ANOVA with Tukey–Kramer's post-hoc test. Identical letters within each plot indicate a statistically significant grouping (*P* < 0.05).

resulted in markedly reduced DK1042 colony growth (Fig. 6A). A PCR screen of the main biosynthetic gene for DAPG, *phlD*, revealed that 10 of 11 colony-inhibiting isolates carried *phlD* (Fig. 6b). By contrast, zero of seven non-inhibiting strains tested positive for *phlD*. To determine whether DAPG or any intermediate products from the PhlA−D BGC operon influence the interaction, we cocultured *P. protegens* DTU9.1 and mutants lacking *phlACB* (DAPG⁻) or *phlACBD* (phloroglucinol⁻, monoacetylated phloroglucinol⁻, DAPG⁻, chlorinated phloroglucino⁻, and pyoluteorin⁻) with DK1042 in liquid 1× LBGM and on solid 0.1× LBGM (Fig. 6c). Disrupting the production of DAPG in DTU9.1 markedly

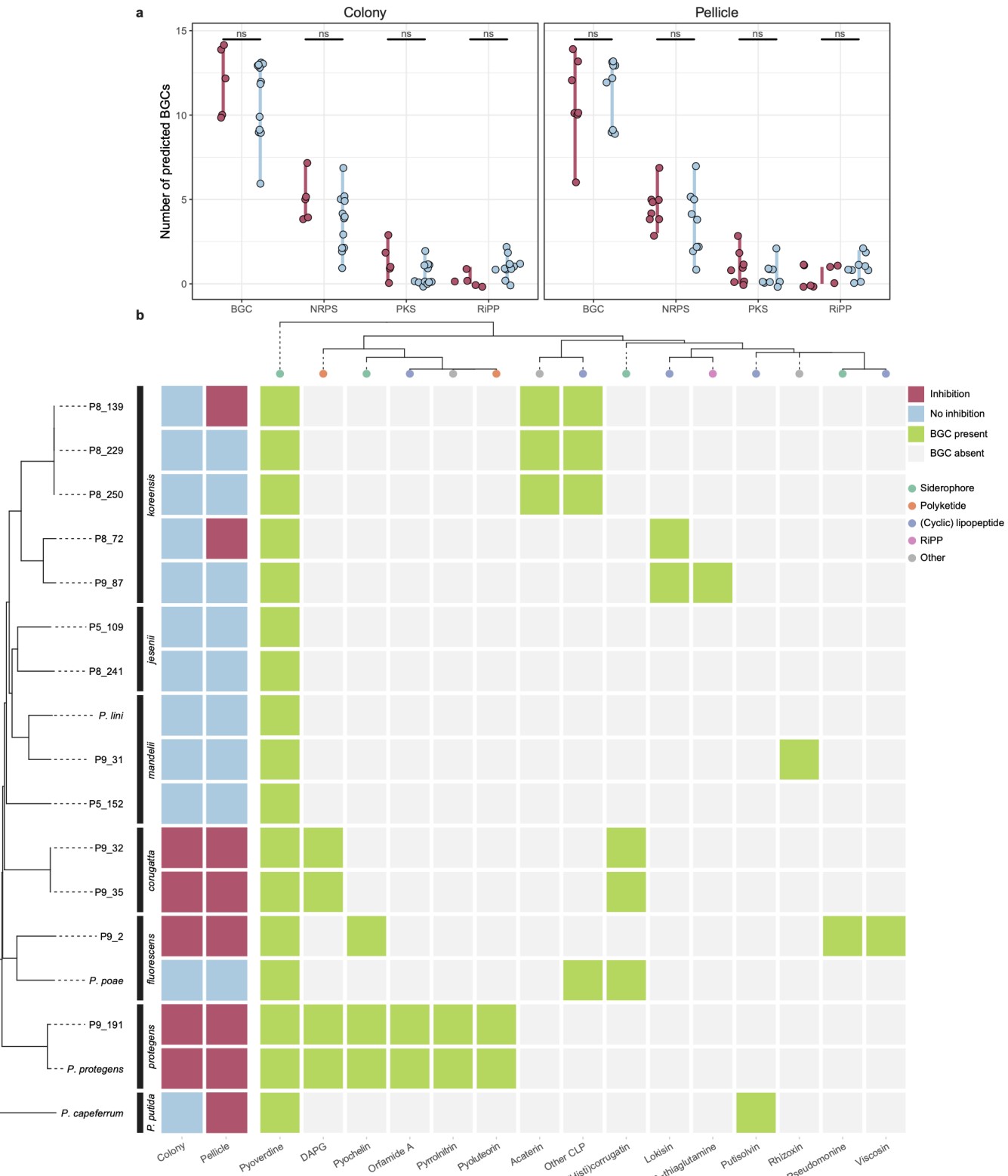

**FIG 4** BGCs (not abundance) predict interaction outcome. Thirteen isolates were whole-genome sequenced and compared with the four reference strains. (a) AntiSmash predictions of BGCs and BGC subgroups grouped by inhibition potential in colony or pellicle cocultures. Statistical tests were pairwise Student's *t*-tests adjusted for multiple testing with the FDR method. ns, not significant (adj. *P* > 0.05). (b) *Pseudomonas*-related natural product machinery encoded within each strain. Strain phylogeny is based on whole-genome sequence identity as described in TYGS (52).

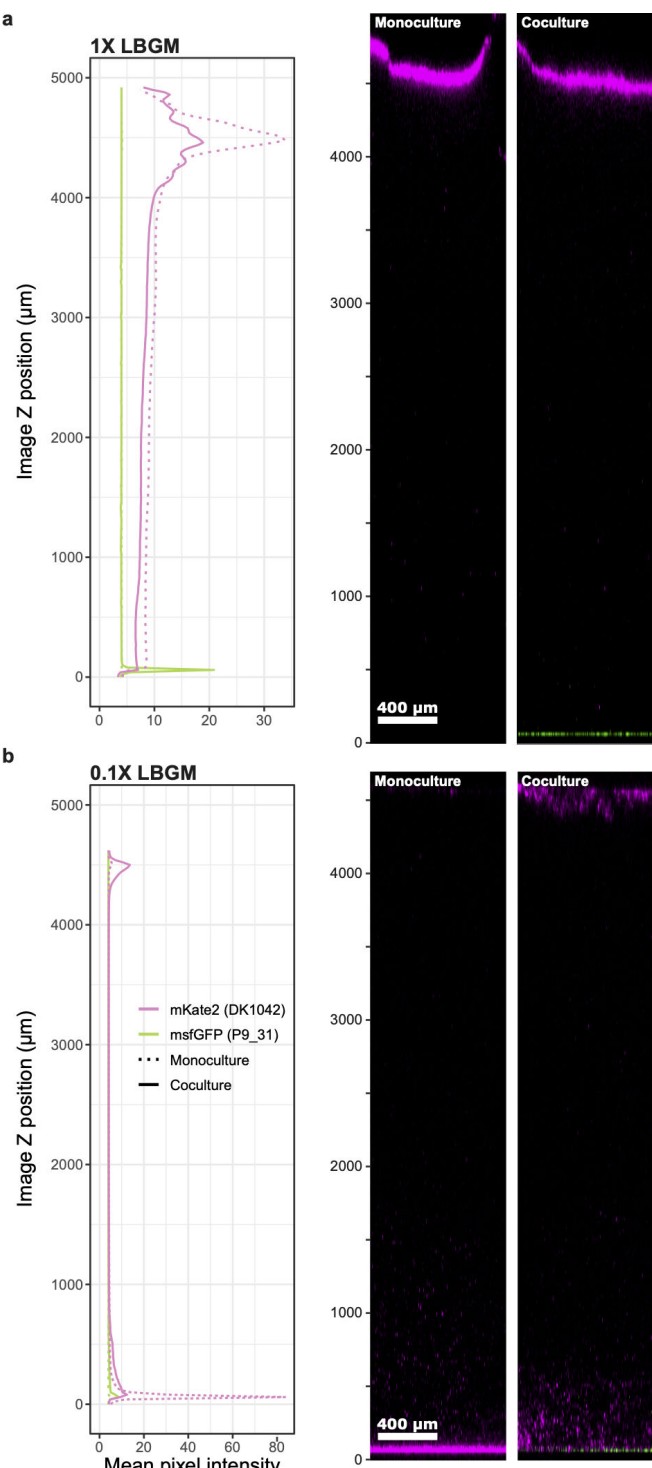

**FIG 5** Non-inhibiting *Pseudomonas* is not present in the pellicle. Coculture of DK1042 (magenta) with non-inhibiting isolate P9_31 (green) in 1× LBGM (a) and 0.1× LBGM (b). Confocal laser scanning microscopy was employed to acquire *Z*-slices every 20 µm spanning the entire depth of a well. Plots show the mean pixel intensity of each strain over the depth of the well. Pixel intensities cannot be compared between 1× LBGM and 0.1× LBGM.

improved DK1042 fitness in colonies but not in pellicles. Thus, DAPG is a strong inhibitor of *B. subtilis* growth in colonies and pellicles, but not the only inhibiting factor.

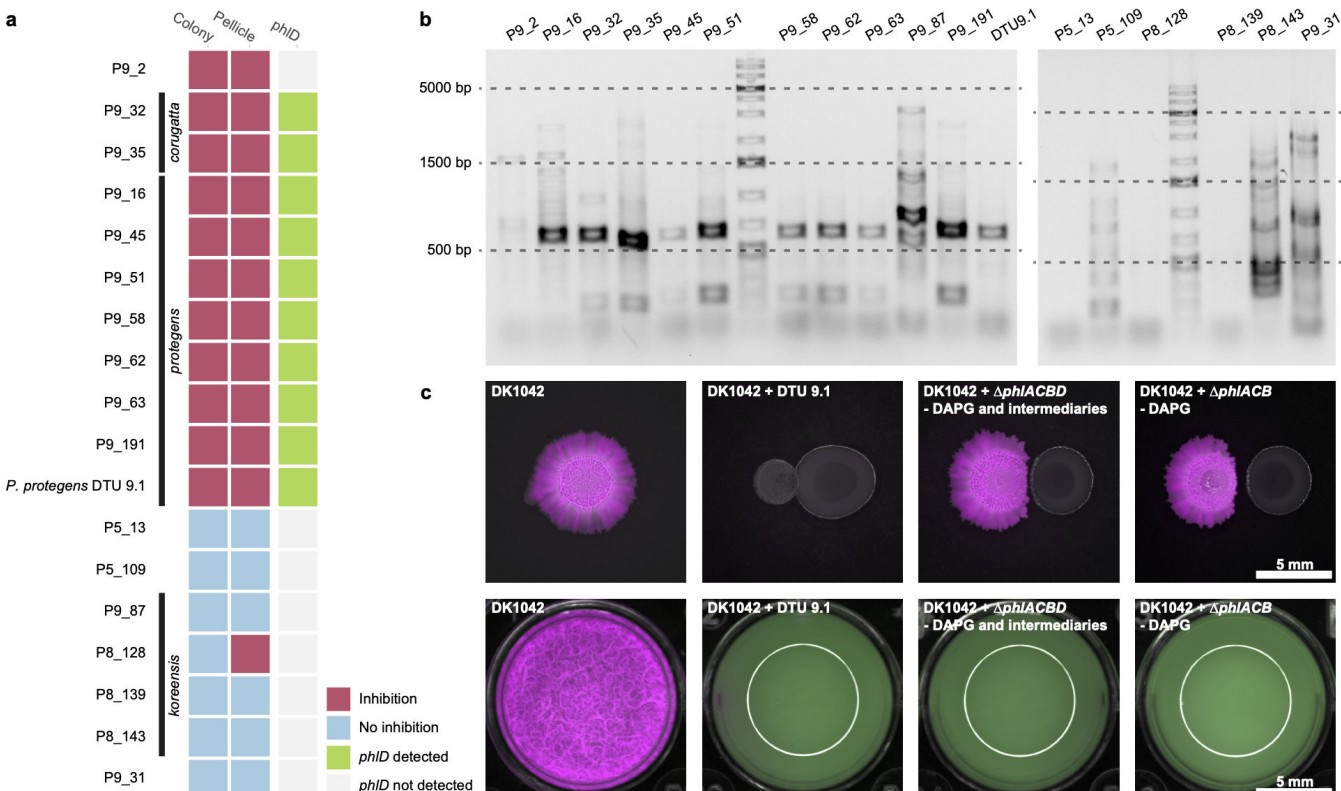

FIG 6  2,4-Diacetylphloroglucinol (DAPG) is a major (but not exclusive) antagonistic metabolite. (a) Inhibition potential of a selection of ifsolates and the presence/absence of *phlD* as determined by PCR (b). *phlD* was amplified with primers yielding a product of ~630 bp. Ladder is GeneRuler 1 kb plus DNA ladder (Thermo Fisher Scientific, Waltham, MA, USA). (c) Cocultures of DK1042 and *P. protegens* DTU 9.1 and *phl*-mutant derivatives of DTU 9.1. Δ*phlACB* cannot produce DAPG. Δ*phlACBD* additionally cannot produce phloroglucinol, monoacetylated phloroglucinol, chlorinated phloroglucinol (PG-Cl/PG-Cl₂), or pyoluteorin.

## DISCUSSION

In this study, we propose a generalization of *B. subtilis-Pseudomonas* interactions based on *Pseudomonas* taxonomy. Our results suggest that fluorescent pseudomonads are very likely to inhibit the *B. subtilis* pellicle, particularly in 0.1× LBGM. Many studies have demonstrated how nutrient sources determine the outcome of pairwise interactions (18, 48, 60, 61). Most of our pairwise interactions were negative, fitting the current paradigm. Large-scale pairwise interaction studies have demonstrated that most pairwise interactions are negative, and that the direction of interaction depends on carbon source complexity (48, 62, 63). It is interesting, however, that such a large proportion of the interactions we tested proved to be negative or neutral. Granted, our setup only shows the interaction outcome of one of the two participants; hence, we cannot differentiate between mutualism (+/+) and parasitism (+/−). Even so, Kehe et al. (48) reported that for 23% of their >180,000 pairwise interactions at least one participant benefitted from coculturing (48), while we report only 2.8% positive interactions, making interactions between *B. subtilis* and fluorescent pseudomonads less likely to be positive compared with average interactions among culturable microorganisms.

This could be due to our decision to coculture under pellicle-inducing conditions. DK1042 proved less durable in liquid broth compared with solid agar. It is likely that both the pseudomonads and DK1042 have different metabolic profiles under the two conditions, and that antagonistic molecules are produced in one setting and not the other. Alternatively, the concentration gradients could be more homogeneous under liquid conditions; for example, DK1042 may be unable to reach adequate cell density for biofilm activation before being growth-inhibited. It is likely that a similar screen performed on solid agar media would result in fewer examples of

*Pseudomonas*-mediated antagonism. Notably, the specific use of strain DK1042 has a limitation, as other strain of *B. subtilis* and other species from the *Bacillus* genus might have competitive advantage against *Pseudomonas* spp. However, the so far described interaction studies between bacilli and pseudomonads do not display species-specific correlations (28). Further experiments are necessary with diverse *Bacillus* species to test the conservation of the here-described correlation between *Pseudomonas* phylogeny and competitive or mutualistic pairwise interaction.

Predicting interactions from phylogeny is arguably a cornerstone of ecology. Charles Darwin proposed what would become the competition-relatedness hypothesis, stating that closely related species are more likely to compete due to niche overlap (64). Indeed, relatedness has consistently correlated with competitiveness, though some find that there is a competitive "peak" at intermediate relatedness (65, 66). Bacilli and pseudomonads are phylogenetically distant, belonging to distinct phyla, though metabolically similar enough that syntropy between isolates has been reported (18). As such, the competition-relatedness hypothesis states that the members of these genera should generally compete. Additionally, these are organisms with large genomes and the potential to produce several antimicrobial compounds. Such organisms cluster into highly competitive communities in metabolic simulations (67). Additionally, the members of *Bacillus* and *Pseudomonas* are frequently co-isolated, suggesting frequent encounters and possible co-evolution. Interactions between *Streptomyces* soil isolates have argued that local evolution is a stronger contributor to interaction outcomes than phylogeny (68, 69). Interactions from a single grain of soil were dramatically different even across isolates with almost identical 16S rDNA sequences, and comparisons between three soil sites found interaction network distributions to differ significantly, again irrespective of phylogenetic distance. In a recent publication, Pomerleau *et al.* (39) demonstrated how adaptive laboratory co-evolution of *B. subtilis* with fluorescent pseudomonads increases *B. subtilis* competitive potential (39). Thus, it is plausible that competitive interactions between bacilli and pseudomonads stem from co-evolution. Predicting interactions from taxonomy requires a certain degree of conservation within taxonomic units. Indeed, with *Pseudomonas* genomes, accessory genes have been found to evolve with core genes, suggesting little horizontal gene transfer, and thus conservation across *Pseudomonas* phylogeny (70, 71). Recently, phylogenetic distance was demonstrated to correlate positively with predictive ability in pairwise interactions, and interactions are conserved within a taxonomic unit but vary between units (60). We propose that *Pseudomonas* constitutes a taxonomic unit with phylogenetically correlative phenotypes, at least in interactions with *Bacillus*. This observed correlation of phylogeny and interaction outcome provides potential for future bioformulations. Both *P. trivialis* and *P. lini* have been implicated in biocontrol and biostimulation (72–74), and their compatibility with *B. subtilis*, therefore, implicates them as candidates for beneficial *Bacillus-Pseudomonas* consortia. Future studies should investigate how mixed cultures function in the rhizosphere and in more diverse communities.

Positive interactions between *Bacillus* and *Pseudomonas* have been reported before, but not with isolates from the fluorescent *Pseudomonas* group. Sun et al. (18) molecularly characterized a mutualistic relationship between *Bacillus velezensis* and *Pseudomonas stutzeri* (now *Stutzerimonas degradans*) (18) and described how *B. velezensis* arrives first and subsequently recruits *Pseudomonas* spp. This interaction gave rise to a mixed biofilm of homogeneously distributed *B. velezensis* and *S. degradans*, which we did not observe. Rather, we observed spatial segregation to the air–liquid interface (DK1042) and the liquid–surface interface (P9_31), though still with enhanced pellicle density in 0.1× LBGM. One could speculate that P9_31 secretes a metabolite that can be distributed throughout the medium and that influences *Bacillus* pellicle formation. However, the generality of such spatial segregation needs to be surveyed.

We found several antagonistic species, and, like others before us (42, 75), experimentally demonstrated the influence of DAPG produced by *Pseudomonas* as a key antagonistic metabolite in neighboring colonies, unlike inhibition by *P. protegens* in pellicle

remains even in the absence of DAPG production. This observation raises the possibility that even if the presence the BGC for DAPG production is a valid genetic marker for inhibition by pseudomonads, it is not the compound that is responsible for the inhibition in pellicle. Additionally, species without the biosynthetic potential to produce DAPG were also characterized as antagonistic, herein the members of the *P. putida* group (*P. entomophila* and *P. capeferrum*) and the *P. fluorescens* subgroup. Previous studies demonstrated how cyclic lipopeptide-producing pseudomonads can inhibit *Bacillus* spp. (76), and given their potential to produce one or more cyclic lipopeptides, the isolates presented here likely share a similar antagonistic property mediated by bioactive specialized metabolites.

Understanding cocultures of *Bacillus* and *Pseudomonas* is pertinent to their applicability in biotechnology. Especially within agriculture, PGPR are being investigated as alternatives to traditional fertilizers and pesticides. While molecular depiction of interaction between *Bacillus* and *Pseudomonas* in the laboratory does not recapitulate the complexity present in the rhizosphere, positive interactions determined *in vitro* might be translated to plant growth promotion of two species consortia (18). Both *B. subtilis* and many fluorescent pseudomonads are characterized as plant growth-promoting agents, and several products based on members of either genus are currently available (77, 78). Interestingly, several studies report having successfully combined bacilli and pseudomonads and achieved some form of synergy (4, 79). For example, the inoculation of a plant growth-promoting *Bacillus* strain can increase the abundance of selected *Pseudomonas* in the rhizosphere (18, 80), which might offer an approach to identify positively interacting pairs of isolates from these two genera facilitating a direct agricultural application. Our results suggest that this synergy likely does not stem from increased *Bacillus* growth. This does not contradict the aforementioned studies, which do not report on the synergistic growth of members, but on outcomes of other parameters (often plant growth with or without stress). One study even reported biocontrol synergy from a mixture of *B. subtilis* and *P. protegens* (81), which suggests either that continued growth of both participants is not required for synergy, or that higher-order interactions abolish DAPG-mediated antagonism of *B. subtilis*. Alternatively, beneficial influence toward a host by isolates from these two bacterial genera might surpasses their negative interactions and does not depend on mutualism.

Both species may not be required to be present at the same time. If the hypothesis proposed by Sun and colleagues that *Bacillus* recruits *Pseudomonas* to the rhizosphere is correct, it is possible that *Pseudomonas* is the effector of biocontrol, eradicating *Bacillus* in the process. However, the fact that many studies have isolated the members of both genera from the rhizosphere provides evidence supporting coexistence of the two. Our pairwise interactions then suggest that other factors underpin *Bacillus* and *Pseudomonas* compatibility.

Although future studies are needed to probe the entire breadth of *Pseudomonas* phylogeny and how it enables interaction predictions, our work demonstrates the relevance in doing so.

## MATERIALS AND METHODS

### Culturing and genetic modification

A library of 720 soil isolates was acquired from soil samples taken from pristine grassland in Dyrehaven Lyngby, Denmark, by plating solubilized soil on King's B agar (KB; 20 g/L peptone, 1% [vol/vol] glycerol, 8.1 mmol/L $K_2HPO_4$, 6.08 mmol/L $MgSO_4 \cdot 7H_2O$) supplemented with 40 µg/mL ampicillin, 13 µg/mL chloramphenicol and 100 µg/mL cycloheximide. Plates were incubated at 30°C for 5 days, and colonies were assessed for fluorescence and placed into lysogeny broth (LB; Lennox, Carl Roth, Karlsruhe, Germany) in 96-well microtiter plates. We labelled the isolation sites as P5 (*n* = 237 isolates), P8 (*n* =

279), and P9 ($n$ = 224), where P5 and P9 came from short grass, while P8 came from long grass.

DK1042 and soil isolates were routinely cultured in tryptone soy broth (CASO broth; Sigma-Aldrich, Darmstadt, Germany), LB, LB supplemented with 1% glycerol and 0.1 mM MnCl$_2$ (1× LBGM), and a 10× dilution of LBGM (0.1× LBGM) at 30°C. Solid media were supplemented with 1.5% (wt/vol) agar. Antibiotics were added as appropriate in the following final concentrations: gentamycin (Gm) 50 µg/mL, ampicillin (Amp) 100 µg/mL, chloramphenicol (Cm) 10 µg/mL, nalidixic acid (NalAc) 20 µg/mL.

*Pseudomonas* soil isolates were fluorescently tagged by inserting constitutively expressed msfGFP into the attTn7 site as described previously (82).

## High-content screening

Precultures of isolates and *B. subtilis* DK1042 Phy-mKate2 were mixed in equal volumes in 96-well imaging microtiter plates (PerkinElmer, Waltham, MA, USA) in TSB or 0.1× LBGM (Fig. 1a). Isolates were adjusted to a final dilution of 100-fold and DK1042 to a final OD$_{600}$ = 0.01. One column contained DK1042 monocultures and one column non-inoculated medium (blank control). Pellicles were incubated at 30°C for 24 h, before removing the supernatant underneath the pellicle and scanning the plate in an Opera QEHS high-content screening microscope (PerkinElmer) equipped with a UAPO20 × W3/340 objective with NA = 0.7.

Focal height was adjusted using monoculture samples, and each plate was scanned by imaging four random locations per well, acquiring 21 *Z*-slices in increments of 2.0 µm from the bottom. Samples were excited with a 561 nm laser collecting emission light through a 690/70 nm filter for mKate2 fluorescence. Laser power was set to 100 µW, and samples were excited for 800 ms.

Opera flex-files were imported into FIJI (2.1.0/1.53f51) (83) using BioFormats and segmented by applying a 5 × 5 mean convolution kernel to remove noise, a 3D median filter with radius *Z* = 2.0 µm to remove single cells or objects only present in one slice, and applying the built-in ImageJ Remove Background function with a rolling ball size of 100 px. Images were then thresholded using the MaxEnthropy algorithm (84) based on the pixel intensities in the entire volume (Fig. S1).

Biovolume was calculated using BiofilmQ (85) by importing the segmented images and sectioning the segmentation into 8 µm cubes. Objects smaller than 8 µm$^3$ were filtered out, and the total biovolume from each image stack was determined.

Subsequent analysis was carried out in Rstudio (2022.02.3-b492) (86) within R (4.1.1) (87) with the Tidyverse framework (1.3.1) (88). Log$_2$(FoldChange) was calculated between the coculture biovolume of each isolate and the monoculture biovolume in the corresponding plate. Assuming that each participant in a biofilm is theoretically able to occupy 1/$k$ of the space (where $k$ is the number of participants), we divided the biovolume from the monoculture by $k$ to take this lack of space into account.

Isolates were divided into three categories (neutral, negative, or positive) based on the median and interquartile range of three biological replicates (Fig. 2a).

## *rpoD*-targeted amplicon sequencing

To determine the taxonomic composition of the screen output, we employed a targeted amplicon sequencing approach using the species-specific gene *rpoD* as previously demonstrated (89).

Precultures of isolates were adjusted to OD = 0.5 and pooled by equal volume according to their category. Genomic DNA was extracted twice from 100 µL of each pool using a Bacterial & Yeast Genomic DNA Purification Kit (EURx, Gdańsk, PL) following the manufacturer's instructions, yielding 1,000–1,500 ng of DNA in 50 µL. DNA was also extracted from a pool of all 720 isolates and as a negative control, nuclease-free H$_2$O was used in place of a bacterial pool. From each DNA extraction, 10 ng was used as template

in two PCR amplification experiments using TEMPase hot start polymerase (Ampliqon, Odense, DK) and barcoded primer pairs (Table S1), resulting in two PCR mixtures from each of two genomic DNA extractions. A final 25 µL PCR experiment consisted of TEMPase mastermix (1×), forward primer (320 nmol/L), reverse primer (320 nmol/L), MgCl$_2$ (1.75 mmol/L), template gDNA, and nuclease-free H$_2$O. The reaction was initiated with 15 min at 95°C to denature the DNA and activate the TEMPase polymerase, followed by 35 cycles of 30 s denaturation (95°C), 30 s annealing (53°C), 30 s elongation (70°C), and a final 5 min elongation step at 70°C. Read length was assessed by DNA agarose gel electrophoresis (1% [wt/vol]), and amplicons were purified from the PCR mix using a NucleoSpin Gel and PCR Clean-up kit (Macherey-Nagel, Düren, DE) following manufacturer's instructions.

DNA purities and concentrations were assessed using a Denovix spectrophotometer (DeNovix, Wilmington, DE, USA) and a Qubit 2.0 fluorimeter (Thermo Fisher Scientific) and a Qubit High Sensitivity kit, respectively. Optical density ratios (260/280 and 260/230) measured 2.0 ± 0.20 and concentrations ranged from 20 to 68 ng/µL. Amplicon DNA (360 ng) from each sample was pooled to a total of 10 µg DNA and sent to Seqomics Biotechnology Ltd. for sequencing on a MiSeq platform (Illumina, San Diego, CA, USA) using a MiSeq Reagent Kit v3 (600-cycle).

The resulting reads were quality-checked and demultiplexed using CutAdapt V4.0 (90). FastP V0.23.2 (91) was used with default settings for quality filtering. Mapping and post-mapping filtering were performed using the bowtier.sh script from (89). In brief, paired reads were aligned with bowtie2 to a custom database of 160 *rpoD* genes. The resulting SAM-file was then filtered for only corresponding paired reads mapped with a quality >10 using samtools. In Rstudio, a permutational multivariate ANOVA was performed to test for between-sample clustering. Amplicon reads were normalized with DESeq2 (92) before calculating relative abundances. Differential abundance analysis was carried out using ANCOM-BC (93) with uncorrected read counts, and multiple testing was corrected with the false discovery rate (FDR) method (94).

In addition, Sanger sequencing of the *rpoD* gene was used for individual routine taxonomic identification of isolates described in Table S1 with primers PsEG30F and PsEG790R (89).

## Pairwise interactions

On agar, DK1042 *amyE*::P$_{hyperspank}$-*mKate2 sacA*::P$_{eps}$-*gfp* and a candidate isolate were spotted 5.0 mm apart on agar surfaces using 2 µL of culture at an OD$_{600}$ of 1.0. Before spotting, plates were dried for 30 min in a lateral flow hood and then incubated at 30°C for 72 h.

In broth, the DK1042 *amyE*::P$_{hyperspank}$-*mKate2 sacA*::P$_{eps}$-*gfp* strain and a candidate isolate were mixed 1:1 volumetrically in 1 mL liquid LBGM in 24-well microtiter plates at a final OD$_{600}$ of 0.01 and incubated for 24 h at 30°C.

Example *Pseudomonas* strains were *P. lini* 1.6, *P. poae* DSM 14936 (95), *Pseudomonas kermanshahensis* F8 (previously *P. capeferrum*) (58) and *P. protegens* DTU 9.1 (58).

## Stereomicroscopy

Colonies and pellicles were imaged with a Carl Zeiss Axio Zoom.V16 stereomicroscope (Carl Zeiss, Oberkochen, Baden-Württemberg, Germany) equipped with a CL 9000 LED light source (Carl Zeiss) and an AxioCam 503 monochromatic camera (Carl Zeiss). The stereoscope was equipped with a PlanApo Z 0.5×/0.125 FWD 114 mm, and the filter sets 38 HE enhanced GFP (eGFP) (ex: 470/40, em: 525/50) and 63 HE mRFP (ex: 572/25, em: 629/62). Exposure time was optimized for contrast but kept constant under identical conditions (i.e., medium type or biofilm type).

Image processing and analysis were performed in FIJI. Contrast in fluorescence channels was adjusted identically on a linear scale to allow for visual comparisons between images with equal exposure time. Reporter fluorescence intensity was

measured by segmenting the colony of interest with the Triangle thresholding algorithm (96) based on mKate2 signal and measuring relative eGFP intensity per area within the resulting region of interest outlining the entire colony. For pellicles, circles were manually fitted to include only the well. Standard deviation of the pixel intensity was measured and reported as a proxy for pellicle wrinkles.

## Confocal microscopy

DK1042 was cultured with isolate P9_31 in 1× LBGM or 0.1× LBGM in 24-well imaging microtiter plates (PerkinElmer). Wells were imaged on a Leica SP8 confocal microscope (Leica Microsystems, Wetzlar, Germany) equipped with an HC PL Fluotar 10x/0.30 air objective and lasers exciting at 488 and 552 nm. Photomultiplier tubes were adjusted to acquire photons at wavelengths of 493–581 (msfGFP) and 586–779 nm, and gain was adjusted for optimal contrast, but kept constant for each media type. Images were acquired with eight bits and size 512 px × 512 px × 247 px (XYZ – voxel size: 1.78 µm × 1.78 µm × 20.00 µm) averaging over four lines.

## Genome mining

Whole-genome sequencing was performed as described previously (82). Assembled genomes were annotated with Bakta (V1.6.1) (97), and a BGC presence/absence matrix was created with antiSMASH (V7.0.0) (98) using its relaxed mode performing Known-ClusterBlast, ClusterBlast, SubClusterBlast, and MIBiG cluster comparison, and ActiveSiteFinder, RREFinder, and transcription factor binding site (TFBS) analysis. Results were manually curated using the *Pseudomonas* Genome Database (99) as reference. PCR screening for *phlD* was performed with primers B2BF (5′-ACCCACCGCAGCATCGTTTATG AGC) and BPR4 (5′-CCGCCGGTATGGAAGATGAAAAAGTC), yielding a 630 bp product.

## ACKNOWLEDGMENTS

This project was funded by a DTU Alliance Strategic Partnership PhD fellowship, by the Danish National Research Foundation (DNRF137) for the Center for Microbial Secondary Metabolites, and the Novo Nordisk Foundation within the INTERACT project of the Collaborative Crop Resiliency Program (NNF19SA0059360), and the "Imaging microbial language in bio-control (IMLiB)" infrastructure grant (NNF19OC0055625).

## AUTHOR AFFILIATIONS

[1]Bacterial Interactions and Evolution Group, DTU Bioengineering, Technical University of Denmark, Kongens Lyngby, Denmark
[2]Microbiome Interactions and Engineering, DTU Bioengineering, Technical University of Denmark, Kongens Lyngby, Denmark
[3]Institute of Plant Biology, Biological Research Center, ELKH, Szeged, Hungary
[4]Institute of Biology Leiden, Leiden University, Leiden, the Netherlands

## AUTHOR ORCIDs

Mark Lyng http://orcid.org/0000-0003-2448-1729
Morten L. Hansen http://orcid.org/0000-0003-3927-2751
Lars Jelsbak http://orcid.org/0000-0002-5759-9769
Gergely Maróti http://orcid.org/0000-0002-3705-0461
Ákos T. Kovács http://orcid.org/0000-0002-4465-1636

## FUNDING

| Funder | Grant(s) | Author(s) |
| --- | --- | --- |
| Danmarks Grundforskningsfond (DNRF) | DNRF137 | Ákos T. Kovács |

| Funder | Grant(s) | Author(s) |
|--------|----------|-----------|
| Novo Nordisk Fonden (NNF) | NNF19SA0059360 | Ákos T. Kovács |
| Novo Nordisk Fonden (NNF) | NNF19OC0055625 | Ákos T. Kovács |
| Technical University of Denmark | PhD fellowship | Mark Lyng |
| | | Ákos T. Kovács |

## AUTHOR CONTRIBUTIONS

Mark Lyng, Conceptualization, Formal analysis, Investigation, Methodology, Software, Visualization, Writing – original draft | Birta Þórisdóttir, Investigation | Sigrún H. Sveinsdóttir, Investigation | Morten L. Hansen, Resources, Writing – review and editing | Lars Jelsbak, Resources, Writing – review and editing | Gergely Maróti, Methodology, Resources, Writing – review and editing | Ákos T. Kovács, Conceptualization, Funding acquisition, Resources, Supervision, Writing – review and editing

## DATA AVAILABILITY

Analysis scripts and processed data have been deposited at Github (https://github.com/marklyng/screen_repository). *rpoD* amplicon sequencing data have been deposited at the Sequence Read Archive under BioProject ID PRJNA985909.

## ADDITIONAL FILES

The following material is available online.

### Supplemental Material

**Data Set S1 (mSystems00212-24-s0001.csv).** Summarized biofilm biovolumes.
**Supplemental figures and table (mSystems00212-24-s0002.pdf).** Fig. S1 and S2; Table S1.

### Open Peer Review

**PEER REVIEW HISTORY (review-history.pdf).** An accounting of the reviewer comments and feedback.

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
