## [Reviewer comments · mSystems]

Taxonomy of *Pseudomonas* spp determines interactions with *Bacillus subtilis*

Mark Lyng, Birta Þórisdóttir, Sigrún Hrefna Sveinsdóttir, Morten Hansen, Lars Jelsbak, Gergely Maróti, and Ákos Kovács

Corresponding Author(s): Ákos Kovács, Universiteit Leiden

Review Timeline:

Submission Date:	February 13, 2024
Editorial Decision:	April 17, 2024
Revision Received:	July 15, 2024
Editorial Decision:	August 9, 2024
Revision Received:	August 12, 2024
Accepted:	August 16, 2024

Editor: Michela Gambino

Reviewer(s): The reviewers have opted to remain anonymous.

Transaction Report:

DOI: <https://doi.org/10.1128/msystems.00212-24>

Re: mSystems00212-24 (Taxonomy of *Pseudomonas* spp determines interactions with *Bacillus subtilis*)

Dear Prof. Ákos T. Kovács:

Revision Guidelines

Sincerely,
Michela Gambino
Editor
mSystems

Reviewer #1 (Comments for the Author):

This study takes a high throughput approach to measuring the impacts of diverse pseudomonads on *Bacillus subtilis* growth and biofilm formation. The rationale is that these taxa regularly interact in plant-associated rhizosphere communities and these interactions may be competitive or positive, but are not well understood. The study finds some intriguing trends based *Pseudomonas* taxonomy and gene context.

Overall the paper was concise and fun to read, and the results are clearly explained and robustly considered. I have some

general comments on the framing of the work, which could have been more clear at times. I do feel that some of the conclusions are overstated and that the discussion is a bit speculative. My main concerns are: 1) The ecological relevance of the phenotypes being measured (biofilm formation in media, especially liquid) in nature was not discussed. Are these assays measuring something that is relevant in the field? 2) Some conclusions feel overly generalized given that *Pseudomonas* is a very diverse genus. Specific comments below:

Clarity of background and rationale:

The framing of the abstract and importance sections was fairly different from the introduction of the paper, which made it a bit jarring when the introduction focused mainly on plant beneficial pseudomonads. I think it would be easier to follow if the abstract also worked in the point about rhizosphere interactions.

The rationale for the different media types used was not well explained and it took me a bit to figure out that there was an expectation about biofilm formation in one type of media. More generally, the introduction would benefit from more information about *Bacillus* biology. The ecological relevance of pellicles and biofilms is not discussed, despite the set up of the paper being about the importance of ecological interactions with these bacteria in nature. The use of high throughput lab assays makes the experiments unavoidably unnatural. I think this is fine, obviously there is a huge benefit in being able to screen a diversity of strains. But there was no mention about whether the traits being measured would matter in soil or in association with a plant

The focus on fluorescence felt odd to me since fluorescent molecules (pyoverdine or pyochelin) were not predictive of outcomes. This stood out in the discussion where the authors referenced this being a novel study because interactions of fluorescent pseudomonads and *Bacillus* had not been previously studied. I think the strength of the study is from the diversity of strains used, the fluorescence doesn't seem critical.

Strength of conclusions:

The discussion of BGCs at the end of the introduction (lines 80-83) felt overstated to me. BGC content generally didn't predict outcomes but here the manuscript states "We found that *Pseudomonas* antagonism towards DK1042 is mainly due to the presence of specific biosynthetic gene clusters (BGCs)". This could be rephrased to make the point about DAPG synthesis but treat the conclusions about other BGCs more fairly.

The discussion feels a bit speculative without clear conclusions. The point about *Bacillus* interactions with *Pseudomonas* taxa being "highly predictable" feels overstated. The authors found an enrichment of certain *Pseudomonas* taxa associated with negative or positive outcomes, but this is not the same as saying that a taxonomic identity will be highly predictive of interactions. This conclusion seems to be based on outcomes being less strain specific than in work on *Streptomyces* interactions, but this was not compelling to me that interactions with *Pseudomonas* would be predictable. Furthermore, the outcomes being measured a fairly coarse and within a specific laboratory set up. Most of these were neutral with respect to the traits being measured, but that leaves a lot of space for more complex interactions in nature.

Minor points:

I would have liked more discussion about DAPG, how common is it, when do bacteria make it?

Line 229 - I think the authors mean phenotypic predictability? Or phylogenetically predictable phenotypes.

Reviewer #3 (Comments for the Author):

Members of the genus *Pseudomonas* and *Bacillus* are common in soil habitats. Both taxa have been associated with plant-beneficial effects. For that reason, there is great interest in studying both ecological and mechanistic aspects of interaction patterns between the two taxa. In their paper, Lyng et al explore the interactions between a focal *B. subtilis* strain and 720 *Pseudomonas* soil isolates. They conducted a high-throughput screen in which they quantified pellicle biofilm formation by *B. subtilis* when co-cultured individually with the 720 *Pseudomonas* strains. They found that many of *Pseudomonas* isolates inhibited *B. subtilis*. Follow-up experiments revealed that inhibitory effects were associated with *Pseudomonas* species classification and more pronounced during pellicle than colony formation. The authors sequenced the genomes of representative *Pseudomonas* isolates to explore the mechanistic basis of *B. subtilis* inhibition. They found that the presence of the DAPG synthesis cluster correlated with inhibitory effects but that not all inhibitory effects can be linked to DAPG.

Overall, this is a solid paper that reports interesting results. The high-throughput screen is remarkable and one of the strongest aspects of the paper. It unravels the large diversity of possible interactions. Another important insight is that ecological interactions can vary as a function of medium composition and culturing type (pellicle vs. colony mode of growth). The

mechanistic parts of the paper are also important but do not manage to conclusively show how *Pseudomonas* inhibits *Bacillus*.

Major comments:

(1) Using 720 *Pseudomonas* strains is powerful to capture the diversity of *Pseudomonas* behaviour. Using a single *Bacillus* strain however limits interpretations about the diversity of *Pseudomonas*-*Bacillus* interactions. This is a limitation of the study design and must be discussed carefully. The authors predominantly found negative and neutral interactions with *B. subtilis* DK1041. Is this a feature of DK1041 because it is not very competitive? Or does it represent a general pattern applicable to many *Bacillus* strains? Could there be specific *Bacillus* strains that interact more often positively with *Pseudomonas* strains? A clear justification for the study design and a concise discussion on the above scenarios must be provided.

(2) Taxonomic classification. It was unclear to me what procedure was used to allocate strains to species groups. Was a threshold identity (e.g. 97%) used as it is common for 16S rRNA amplicon sequencing? The authors state that "concordant pairs" were searched for. Does concordant stand for 100% identity? Finally, the authors say that Sanger sequencing of *rpoD* was used in addition to amplicon sequencing (lines 353+354). But this was clearly not done for all 720 isolates. Why was this done and for how many strains? Some clarifications are clearly needed.

(3) Figure S3 seems relevant for the paper. It statistically shows the link between sequence identity/variation and interaction patterns. This figure is a key result and should be integrated into Fig. 2. Instead, the current panels 2a and 2b could be reduced or moved to the supplements.

(4) The spatial segregation phenomenon shown in Fig. 5 is interesting. However, it is unclear whether this is a general *Pseudomonas*-*Bacillus* interaction pattern or a peculiarity of P9_31. I recommend to down-tone this finding. It is an interesting but an anecdotal observation and should be presented in this context.

(5) The association between presence/absence of the *phlD* gene (part of the DAPG) cluster in *Pseudomonas* and the inhibition of *Bacillus* (Fig. 6a) is remarkable. However, the experimental results with the *phlACBD* mutant are less clear. The initial screen (Fig. 3) shows that *Pseudomonas* inhibits *Bacillus* predominantly in pellicles but not in colonies. In contrast, the experiments with the *phl*-mutant revealed that DAPG is powerful in inhibiting *Bacillus* in colonies but not in pellicles. Thus, the two results do not match and suggest that DAPG is not responsible for the observed pellicle inhibition. These seemingly contradictory results must be discussed carefully. They do not question the value of this work but show how hard it is to identify the mechanisms involved. Moreover, the results suggest that the presence of DAPG is a valid genetic marker for inhibition, but it is not the compound that is responsible for the inhibition. My take on this is that there is a second (undetected) BGC that is perhaps genetically linked to DAPG that is responsible for the inhibition.

(6) The discussion is currently the weakest part of the paper. A more balanced discussion on the actual findings and their implications are required. Above, I have already highlighted a number of key discussion points. Moreover, it is essential to discuss how laboratory experiments can help to understand interactions in complex environments. For example, is the observed *Pseudomonas* competitiveness a consequence of the lab environment or can we expect similar patterns under natural conditions? How representative is the use of a single *Bacillus* strain? How would one now screen and select particularly beneficial species pairs for applications?

Further comments:

(7) Line 19: What exactly is meant by "pairwise compatibility"? Does this term refer to co-existence or positive effects? Some clarifications are needed.

(8) Line 26: The spatial segregation was shown for a single case. It should not be sold as a general pattern. This aspect should either be clarified or removed from the abstract.

(9) Line 53: Here, the authors use the term "symbiosis". Later in the paper, they refer to "mutualism". Both terms are used to describe (+/+) interactions. I suggest to stick to one of the two terms and it seems that "mutualism" is more widely used in this microbiology.

(10) Line 143: I recommend to define BGC once more in the results. Although introduced earlier, readers (including myself) might have forgotten what BGC means at this stage.

(11) Lines 258-262: Another (and perhaps more elegant) way to phrase this is that no mutualism between *Bacillus* and *Pseudomonas* is required for beneficial effects towards third parties. More generally, two species could keep each other in check through competition and this could be good for a host.

Reviewer #1 (Comments for the Author):

This study takes a high throughput approach to measuring the impacts of diverse pseudomonads on *Bacillus subtilis* growth and biofilm formation. The rationale is that these taxa regularly interact in plant-associated rhizosphere communities and these interactions may be competitive or positive, but are not well understood. The study finds some intriguing trends based on *Pseudomonas* taxonomy and gene context.

Overall the paper was concise and fun to read, and the results are clearly explained and robustly considered. I have some general comments on the framing of the work, which could have been more clear at times. I do feel that some of the conclusions are overstated and that the discussion is a bit speculative. My main concerns are: 1) The ecological relevance of the phenotypes being measured (biofilm formation in media, especially liquid) in nature was not discussed. Are these assays measuring something that is relevant in the field? 2) Some conclusions feel overly generalized given that *Pseudomonas* is a very diverse genus. Specific comments below.

We thank the Reviewer for their positive assessments of our work; we have followed the below suggestions to adjust the discussions, so overstates are avoided in the revised version.

Clarity of background and rationale:

The framing of the abstract and importance sections was fairly different from the introduction of the paper, which made it a bit jarring when the introduction focused mainly on plant beneficial pseudomonads. I think it would be easier to follow if the abstract also worked in the point about rhizosphere interactions.

We appreciate this suggestion and to create a better parallelism of this topic in the abstract and introduction. We have indicated now in the abstract the relevance of interaction in the rhizosphere.

The rationale for the different media types used was not well explained and it took me a bit to figure out that there was an expectation about biofilm formation in one type of media. More generally, the introduction would benefit from more information about *Bacillus* biology. The ecological relevance of pellicles and biofilms is not discussed, despite the set up of the paper being about the importance of ecological interactions with these bacteria in nature. The use of high throughput lab assays makes the experiments unavoidably unnatural. I think this is fine, obviously there is a huge benefit in being able to screen a diversity of strains. But there was no mention about whether the traits being measured would matter in soil or in association with a plant.

*We have now introduced previous work that demonstrated correlation of in vitro biofilm formation of *B. subtilis* with plant protection and rhizosphere colonization. In addition, we have included information early in the results section on the use of different media for testing biofilm formation of *B. subtilis*.*

The focus on fluorescence felt odd to me since fluorescent molecules (pyoverdine or pyochelin) were not predictive of outcomes. This stood out in the discussion where the authors referenced this being a novel study because interactions of fluorescent pseudomonads and *Bacillus* had not been previously studied. I think the strength of the study is from the diversity of strains used, the fluorescence doesn't seem critical.

Fluorescent Pseudomonas is generally referring to soil-dwelling Pseudomonas, which are being isolated based on their auto fluorescence, as reviewer also mentions, due to the fluorescent molecules (pyoverdine or pyochelin). The Pseudomonas library that we have used in this study has been obtained by isolating Pseudomonas specifically on KB agar medium, which enables the enrichment and identification of fluorescent Pseudomonas. This focus aims to discard other Pseudomonas species, including Pseudomonas aeruginosa, which is not targeted here. The Pseudomonas field generally use this term, fluorescent Pseudomonas, when focusing on soil derived Pseudomonas isolates.

Strength of conclusions:

The discussion of BGCs at the end of the introduction (lines 80-83) felt overstated to me. BGC content generally didn't predict outcomes but here the manuscript states "We found that Pseudomonas antagonism towards DK1042 is mainly due to the presence of specific biosynthetic gene clusters (BGCs)". This could be rephrased to make the point about DAPG synthesis but treat the conclusions about other BGCs more fairly.

We have adapted the conclusion at the end of the introduction as the Reviewer suggested referring to DAPG instead of generally to BGCs.

The discussion feels a bit speculative without clear conclusions. The point about Bacillus interactions with Pseudomonas taxa being "highly predictable" feels overstated. The authors found an enrichment of certain Pseudomonas taxa associated with negative or positive outcomes, but this is not the same as saying that a taxonomic identity will be highly predictive of interactions. This conclusion seems to be based on outcomes being less strain specific than in work on Streptomyces interactions, but this was not compelling to me that interactions with Pseudomonas would be predictable. Furthermore, the outcomes being measured a fairly coarse and within a specific laboratory set up. Most of these were neutral with respect to the traits being measured, but that leaves a lot of space for more complex interactions in nature.

We agree with the Reviewer. We have removed any claim related to predictability of interaction, and we now refer to the correlation observed between specific phylogenetic groups and inhibition or reduction of B. subtilis biofilm formation in the discussion.

Minor points:

I would have liked more discussion about DAPG, how common is it, when do bacteria make it?

We now introduce relevant information of DAPG producing Pseudomonas species and we refer to previous works on the regulation of its BGC.

Line 229 - I think the authors mean phenotypic predictability? Or phylogenetically predictable phenotypes.

We thank for this suggestion. We have corrected the text accordingly.

Reviewer #3 (Comments for the Author):

Members of the genus Pseudomonas and Bacillus are common in soil habitats. Both taxa have been associated with plant-beneficial effects. For that reason, there is great interest in studying both ecological and mechanistic aspects of interaction patterns between the two taxa. In their paper, Lyng et al explore the interactions between a focal B. subtilis strain and 720 Pseudomonas

soil isolates. They conducted a high-throughput screen in which they quantified pellicle biofilm formation by *B. subtilis* when co-cultured individually with the 720 *Pseudomonas* strains. They found that many of *Pseudomonas* isolates inhibited *B. subtilis*. Follow-up experiments revealed that inhibitory effects were associated with *Pseudomonas* species classification and more pronounced during pellicle than colony formation. The authors sequenced the genomes of representative *Pseudomonas* isolates to explore the mechanistic basis of *B. subtilis* inhibition. They found that the presence of the DAPG synthesis cluster correlated with inhibitory effects but that not all inhibitory effects can be linked to DAPG.

Overall, this is a solid paper that reports interesting results. The high-throughput screen is remarkable and one of the strongest aspects of the paper. It unravels the large diversity of possible interactions. Another important insight is that ecological interactions can vary as a function of medium composition and culturing type (pellicle vs. colony mode of growth). The mechanistic parts of the paper are also important but do not manage to conclusively show how *Pseudomonas* inhibits *Bacillus*.

We appreciate the very positive and supportive assessment by the Reviewer.

Major comments:

(1) Using 720 *Pseudomonas* strains is powerful to capture the diversity of *Pseudomonas* behaviour. Using a single *Bacillus* strain however limits interpretations about the diversity of *Pseudomonas*-*Bacillus* interactions. This is a limitation of the study design and must be discussed carefully. The authors predominantly found negative and neutral interactions with *B. subtilis* DK1041. Is this a feature of DK1041 because it is not very competitive? Or does it represent a general pattern applicable to many *Bacillus* strains? Could there be specific *Bacillus* strains that interact more often positively with *Pseudomonas* strains? A clear justification for the study design and a concise discussion on the above scenarios must be provided.

*We acknowledge these points, and we now extended both the result section and discussions. We explain the use of a single *Bacillus subtilis* strain, which is the most commonly used undomesticated strain used in previous studies, including co-evolution with different *Pseudomonas* strains. Also, we highlight that the *B. subtilis* group forms pellicle, which we aimed to exploit for the high throughput screening described here. In the discussion, we now refer to the summary of previously published examples of interactions between *Bacillus* and *Pseudomonas* genera (Lyng 2023 Trends in Microbiology), which pinpointed no obvious correlation between species of *Bacillus* and mutualism with *Pseudomonas*. Finally, we now discuss that the results are specific for this strain and future work will be necessary to assess the generality of these observations to other *Bacillus* strains and species.*

(2) Taxonomic classification. It was unclear to me what procedure was used to allocate strains to species groups. Was a threshold identity (e.g. 97%) used as it is common for 16S rRNA amplicon sequencing? The authors state that "concordant pairs" were searched for. Does concordant stand for 100% identity? Finally, the authors say that Sanger sequencing of *rpoD* was used in addition to amplicon sequencing (lines 353+354). But this was clearly not done for all 720 isolates. Why was this done and for how many strains? Some clarifications are clearly needed.

We thank the Reviewer for noticing these methodological points, which we now all addressed in the methods description, e.g. "concordant pairs" refer to "corresponding paired reads". We followed the procedures for homology cut off as described by Lauritsen et al. 2021 that used 99%

similarity when mapping against the RDP-II SSU database. We now indicate that the rpoD gene was sequenced for those isolates indicated in Table S1 described by Lyng et al 2024 ISME J.

(3) Figure S3 seems relevant for the paper. It statistically shows the link between sequence identity/variation and interaction patterns. This figure is a key result and should be integrated into Fig. 2. Instead, the current panels 2a and 2b could be reduced or moved to the supplements.

We agree with the Reviewer on the importance of the given data; therefore, we moved Fig S3 graph to Fig 2 as suggested.

(4) The spatial segregation phenomenon shown in Fig. 5 is interesting. However, it is unclear whether this is a general Pseudomonas-Bacillus interaction pattern or a peculiarity of P9_31. I recommend to down-tone this finding. It is an interesting but an anecdotal observation and should be presented in this context.

We have now indicated in the discussion that this is a specific example and additional work needed to demonstrate how broad this phenomenon.

(5) The association between presence/absence of the phlD gene (part of the DAPG) cluster in Pseudomonas and the inhibition of Bacillus (Fig. 6a) is remarkable. However, the experimental results with the phlACBD mutant are less clear. The initial screen (Fig. 3) shows that Pseudomonas inhibits Bacillus predominantly in pellicles but not in colonies. In contrast, the experiments with the phl-mutant revealed that DAPG is powerful in inhibiting Bacillus in colonies but not in pellicles. Thus, the two results do not match and suggest that DAPG is not responsible for the observed pellicle inhibition. These seemingly contradictory results must be discussed carefully. They do not question the value of this work but show how hard it is to identify the mechanisms involved. Moreover, the results suggest that the presence of DAPG is a valid genetic marker for inhibition, but it is not the compound that is responsible for the inhibition. My take on this is that there is a second (undetected) BGC that is perhaps genetically linked to DAPG that is responsible for the inhibition.

We thank the Reviewer for carefully depicting the inhibition profiles by DAPG. We have supplemented the discussion to address this point suggested by the Reviewer.

(6) The discussion is currently the weakest part of the paper. A more balanced discussion on the actual findings and their implications are required. Above, I have already highlighted a number of key discussion points. Moreover, it is essential to discuss how laboratory experiments can help to understand interactions in complex environments. For example, is the observed Pseudomonas competitiveness a consequence of the lab environment or can we expect similar patterns under natural conditions? How representative is the use of a single Bacillus strain? How would one now screen and select particularly beneficial species pairs for applications?

We are grateful for the specific suggestions by the Reviewer above that we have all addressed as indicated. We now indicate that laboratory experiments do not recapitulate the complex environment of the rhizosphere, but examples exist where positive interaction detected in the laboratory between Bacillus and Pseudomonas species translate to plant growth promotion. Finally, we highlight examples where inoculation of single Bacillus strain enhanced the presence of specific Pseudomonas in the plant rhizosphere, that might allow selection of particularly beneficial species pairs for agricultural application.

Further comments:

(7) Line 19: What exactly is meant by "pairwise compatibility"? Does this term refer to co-existence or positive effects? Some clarifications are needed.

As suggested by the reviewer, we refer here to coexistence, which we now indicate in the specific sentence.

(8) Line 26: The spatial segregation was shown for a single case. It should not be sold as a general pattern. This aspect should either be clarified or removed from the abstract.

We agree with the Reviewer's comment, and we now specifically mention the given isolate for spatial segregation.

(9) Line 53: Here, the authors use the term "symbiosis". Later in the paper, they refer to "mutualism". Both terms are used to describe (+/+) interactions. I suggest to stick to one of the two terms and it seems that "mutualism" is more widely used in this microbiology.

That is a valid point, we thank for spotting this. We now use mutualism throughout the manuscript.

(10) Line 143: I recommend to define BGC once more in the results. Although introduced earlier, readers (including myself) might have forgotten what BGC means at this stage.

This has been adjusted.

(11) Lines 258-262: Another (and perhaps more elegant) way to phrase this is that no mutualism between *Bacillus* and *Pseudomonas* is required for beneficial effects towards third parties. More generally, two species could keep each other in check through competition and this could be good for a host.

This is a relevant suggestion and we have now indicated in the discussion.

Re: mSystems00212-24R1 (Taxonomy of *Pseudomonas* spp determines interactions with *Bacillus subtilis*)

Dear Prof. Ákos T. Kovács:

Thank you for the privilege of reviewing your work. Below you will find my comments, instructions from the mSystems editorial office, and the reviewer comments. There are few additional minor details the reviewers spotted that it would good if you could correct.

Revision Guidelines

Sincerely,
Michela Gambino
Editor
mSystems

Reviewer #1 (Comments for the Author):

The authors have sufficiently addressed my comments and the clarity of the manuscript is much improved.

One minor point: I commented previously on the use of the term fluorescent pseudomonad, which the authors explained in their response but didn't address in the manuscript. My point, which I didn't make very clearly before, is that the implications of this

term will not be obvious to a broad readership. We often use terms that have a specific meaning based on context and history, that readers outside of a specific field will not be familiar with. That's fine, so long as you define the term. The manuscript does not define what you mean by the term, so it will only be clear to readers familiar with environmental *Pseudomonas* research. I'd suggest the authors explain what they mean in the paper.

Reviewer #3 (Comments for the Author):

The authors have carefully revised their manuscript. All of my comments have been adequately addressed.

I have spotted a few minor mistakes while rereading the paper.

line 28: "with" should probably be replaced by "which" to turn this into a meaningful sentence.

line 40: change to "whether a majority of ..."

line 130: Fig. S3 has now become Fig. 2d. Reference to this figure needs to be adjusted.

Reviewer #1 (Comments for the Author):

The authors have sufficiently addressed my comments and the clarity of the manuscript is much improved.

> We thank the Reviewer for acknowledging our revisions.

One minor point: I commented previously on the use of the term fluorescent pseudomonad, which the authors explained in their response but didn't address in the manuscript. My point, which I didn't make very clearly before, is that the implications of this term will not be obvious to a broad readership. We often use terms that have a specific meaning based on context and history, that readers outside of a specific field will not be familiar with. That's fine, so long as you define the term. The manuscript does not define what you mean by the term, so it will only be clear to readers familiar with environmental *Pseudomonas* research. I'd suggest the authors explain what they mean in the paper.

We thank for this suggestion. We have now adjusted the text, so fluorescent *Pseudomonas* term is clearly defined. We have now extended the first sentence in the results:

“Relationships between *B. subtilis* and fluorescent pseudomonads (i.e. soil-dwelling *Pseudomonas* sp., that are generally isolated based on their auto fluorescence derived from secondary metabolites, including pyoverdine or pyochelin) range from antagonism to co-existence and, potentially, to synergistic growth, but no systematic investigation of interaction outcomes between the two has been performed.”

Reviewer #3 (Comments for the Author):

The authors have carefully revised their manuscript. All of my comments have been adequately addressed.

I have spotted a few minor mistakes while rereading the paper.

> We thank the Reviewer for spotting these remaining minor mistakes, all have been adjusted.

line 28: "with" should probably be replaced by "which" to turn this into a meaningful sentence.

> Adjusted as requested.

line 40: change to "whether a majority of ..."

> Adjusted as requested.

line 130: Fig. S3 has now become Fig. 2d. Reference to this figure needs to be adjusted.

> Adjusted as requested.

Re: mSystems00212-24R2 (Taxonomy of *Pseudomonas* spp determines interactions with *Bacillus subtilis*)

Dear Prof. Ákos T. Kovács:

Your manuscript has been accepted, and I am forwarding it to the ASM production staff for publication. Your paper will first be checked to make sure all elements meet the technical requirements. ASM staff will contact you if anything needs to be revised before copyediting and production can begin. Otherwise, you will be notified when your proofs are ready to be viewed.

Sincerely,

Michela Gambino
Editor
mSystems